# Automatized analysis of children's exposure to child-directed speech in reschool settings: Validation and application

**Hugo Gonzalez Villasanti**[1]*, **Laura M. Justice**[1,2], **Leidy Johana Chaparro-Moreno**[1,2], **Tzu-Jung Lin**[1,2], **Kelly Purtell**[1,3]

1 Crane Center for Early Childhood Research and Policy, The Ohio State University, Columbus, OH, United States of America, 2 Educational Psychology Program, Department of Educational Studies, The Ohio State University, Columbus, Ohio, United States of America, 3 Human Development and Family Science Program, Department of Human Sciences, The Ohio State University, Columbus, Ohio, United States of America

* gonzalezvillasanti.1@osu.edu

**Data Availability Statement:** All code, validation and test data files are available at https://osf.io/

## Abstract

The present study explored whether a tool for automatic detection and recognition of interactions and child-directed speech (CDS) in preschool classrooms could be developed, validated, and applied to non-coded video recordings representing children's classroom experiences. Using first-person video recordings collected by 13 preschool children during a morning in their classrooms, we extracted high-level audiovisual features from recordings using automatic speech recognition and computer vision services from a cloud computing provider. Using manual coding for interactions and transcriptions of CDS as reference, we trained and tested supervised classifiers and linear mappings to measure five variables of interest. We show that the supervised classifiers trained with speech activity, proximity, and high-level facial features achieve adequate accuracy in detecting interactions. Furthermore, in combination with an automatic speech recognition service, the supervised classifier achieved error rates for CDS measures that are in line with other open-source automatic decoding tools in early childhood settings. Finally, we demonstrate our tool's applicability by using it to automatically code and transcribe children's interactions and CDS exposure vertically within a classroom day (morning to afternoon) and horizontally over time (fall to winter). Developing and scaling tools for automatized capture of children's interactions with others in the preschool classroom, as well as exposure to CDS, may revolutionize scientific efforts to identify precise mechanisms that foster young children's language development.

## Introduction

Language development during the early years of life is driven largely by exposure to the talk of others [1], and linguistic input directed to the child serves to shape the cortical regions of the brain responsible for processing linguistic forms and functions [2]. While there is some controversy as to whether such linguistic input must be directed explicitly to the child, versus overhead by the child but directed to others [3, 4], compelling evidence indicates that child-

m54pc/?view_only=
1ad89d9d285c4cb9aa32141ca10739d5.

**Funding:** The authors received no specific funding for this work.

**Competing interests:** The authors have declared that no competing interests exist.

directed speech (CDS) is an especially important driver of young children's language development [5].

Scientists who study CDS customarily rely on time-intensive, meticulous processes to hand-transcribe and codify characteristics of CDS to which children are exposed; in turn, researchers seek to precisely represent children's early language experiences and use this to understand how CDS influences development and the extent to which CDS exposure varies individually across children [4–7] and over time [8]. However, because of the laborious nature of this work, much of the research on CDS relies on relatively brief recordings of interactions [9] reflecting just a sliver of the CDS children experience over time. Furthermore, human coding inevitably involves biases, and researchers may struggle with inter-rater reliability issues. Establishing automatized systems can facilitate the scientific inquiry of CDS.

To accelerate research on CDS as an important characteristic of young children's proximal caregiving environments, there is a great need to explore whether transcription and codification of CDS can be automatized using emerging technologies. To this end, the purpose of the present work is to present the Classroom Interaction Detection and Recognition (CIDR) system for the automatic transcription and coding of CDS experienced by children in preschool settings, empirically validate the CIDR system, and apply the system to a novel corpus of observational data.

Development and validation of CIDR builds on prior efforts to automatize coding of children's linguistic experiences, most notably through the use of the Language ENvironmental Analysis (LENA) system. Currently marketed as the "industry standard" (lena.org), the system includes both hardware in the form of a wearable recorder that captures child-directed talk and software that processes the recorded talk. LENA analytics can provide estimates for adult word count (AWC), child vocalization count (CVC), and adult-child conversational turns from audio signals. AWC represents the average number of words spoken by adults near a child over an hour, whereas CVC represents the average number of vocalizations produced by a child over an hour, with the metric including words, babbles, and vocalizations [10]. Adult-child conversational turns represents the number of back-and-forth exchanges between the child and any adult per hour.

These automatized data have been deployed for a variety of purposes. For instance, researchers used LENA recordings to identify distinctive features of vocal development in children with autism [11], to explore the linguistic experiences of children with hearing loss in the home environment [12], to examine the effects of peer-to-peer talk in preschool classrooms on children's language growth [13], and to assess the effectiveness of interventions designed to increase parents' talk to their children [14]. In addition, researchers recently have used the AWC data from day-long LENA recordings to link variability in children's language exposure in the home environment to language-related brain structure in terms of neural connectivity [2] and cortical surface area [15].

Despite these and many other important contributions to the literature, there are some limitations to the industry standard for automated capture of children's linguistic experiences. First, the system does not distinguish CDS from non-CDS among adult talk captured in audio recordings. Therefore, researchers concerned with identifying children's exposure to CDS must utilize extensive additional analyses and hand-coding, leading to only modest amounts of audio-recorded talk to be evaluated (e.g., [12]). Second, the system does not provide nuanced measures of linguistic complexity, such as number of different words and mean length of utterance, which represent important aspects of CDS that correlate with children's language development [6]. Third, with the rapid pace of developments and deep-learning applications in the automatic speech recognition (ASR) community, it is unclear whether LENA has adopted these methods, and being a closed-source software hinders the possibility

of adapting the system to custom acoustic environments and new developments in automatic speech recognition [16]. Fourth, recent reviews raise concerns about the validity of the tool [17]. As Cristia and colleagues demonstrate, there have been a number of validity studies examining the performance of LENA relative to hand-coding and transcription in early childhood settings, but many of these do not provide adequate data to compute agreement measures with reference values. While the performance for adult word count estimates was high, the results for child vocalization counts showed a tendency toward underestimation, as well as low correlation with reference values on conversational turn counts. Furthermore, Cristia and colleagues also point out the lack of studies reporting on the accuracy of distinguishing word counts for the focal child from other children in the recording, which is critical when implementing the system in settings with multiple children, such as preschool classrooms. It is important to note that several of the studies reviewed in [17] utilize LENA in settings that do not match LENA's training data, recorded on English-speaking home environments of children ages 1 to 42 months old [18].

To address these limitations, research was commenced to develop an open-source tool that would automatically identify the frequency and key characteristics of CDS experienced by preschool-aged children. A particular interest was developing an approach to capture children's experiences in preschool classrooms, as there is significant interest in understanding children's language-specific experiences in these settings. For instance, researchers have sought to characterize the ways in which teachers talk to children in their preschool classrooms, as this is presumed to be a key driver of early language development [9, 19, 20]. However, most, if not all of this work is focused at the teacher level and does not capture the individual child's experiences. Thus, this study's goal was to develop a system that would represent a given child's exposure to CDS in the classroom setting and depict individual differences among children in this important early experience. Training the system involved using hand-coded reference (ground truth) data generated to examine individual differences in preschool children's CDS exposure in preschool settings from previous research [21]. In order to make the tool as accessible as possible, and without compromising accuracy, cloud-computing multimedia processing services were employed to obtain facial and speech features when adults interacted with and spoke to focal children, which are the inputs to deep-learning classifiers that detect interaction instances.

In the present report, we describe development and training of the system and address the following questions. First, used in combination with speech activity, can facial features predict children's interactions in a preschool classroom? Second, can facial features and speech activity be used to detect and measure CDS exposure in a preschool classroom? Third, are there significant differences in children's CDS exposure across a classroom day (morning to afternoon) and over time (fall to winter)?

## Method

### Participants

This study involved human subjects and was approved by the Ohio State University IRB. Participating teachers and children were drawn from one non-profit child-care center located in an urban neighborhood of a large city. The center serves approximately 100 children from 6 weeks of age to 5 years in seven classrooms operating on a 10-hour basis (7:30 AM to 5:30 PM). For the present study, we recruited one preschool classroom to participate, with the particular classroom recommended by the center principal based on the needs of the study, the anticipated levels of consent from caregivers, teachers' openness to using technology, and other ongoing activities underway in the classroom.

Consent was solicited from all teachers and the primary caregivers of children in the classroom. The three teachers (one master teacher, two assistant teachers) and 19 of the 20 children were consented. For the children, consent could take two forms: consent for the child to wear a head-mounted camera in the classroom, or for the child to be recorded but not wear the camera. Written consent was provided for 15 children to wear the camera, and for 4 additional children to be recorded but not wear the camera; for the latter, consent allowed children to be recorded. For the one child in the classroom for whom no consent was provided, we arranged alternative activities for the child to experience when the recording was taking place.

The teachers were all female and had at least an associate's degree or higher. The children ranged in age from 35 to 58 months at the time of the first observation, and were diverse in terms of race (4 White, 12 African American, 3 un-reported or another race) and socioeconomic status based on maternal education (1 mother did not complete high school, 1 completed high school, 5 completed a certificated training after high school, 4 completed a bachelor's degree, 5 a master's degree, and 2 an educational program beyond the master's degree; 1 un-reported).

## Procedure

The participating children wore head-mounted cameras for around one hour in the morning and one hour in the afternoon at two time-points in the academic year (fall, winter). This study uses existing coding and transcripts of these recordings developed in a prior study investigating preschool children's first-person experiences. Details of the study's design are available in [21], and we briefly review those of relevance to the present study.

**Data source.** During a one-week period in the fall and winter of the academic year, each consented child (focal child) was assigned to wear a wireless head-mounted camera on a randomly assigned day for approximately one hour in the morning and one hour in the afternoon after naptime. Four focal children simultaneously wore the head-mounted camera each day. We followed the same procedure in the fall and winter of the same academic year. Thus, there were potentially up to four recordings per each of the focal children: morning fall (Fall AM), afternoon fall (PM fall), morning winter (Winter AM), and afternoon winter (Winter PM). In actuality, there was variability in the number of children providing data at each time-point due to absences and difficulties with the camera (e.g., the camera fell off a child's head). As a result, video-recordings were available for 13 children for Fall AM, 11 children for Fall PM, 10 for Winter AM, and 11 for Winter PM. Of these, 9 children were recurring on all windows. In addition, the camera was removed for certain activities, such as when children used the restroom.

In this work, we focused on detecting and recognizing focal children's interactions occurring outside whole-group activities, which are more homogeneous in their audiovisual features due to the physical proximity between interacting partners. We believe that automatic transcription of CDS occurring during whole-group activities can be obtained more accurately using audio acquisition devices that are closer to the source, like teacher-worn audio recorders. The recorded classroom tasks outside whole-group activities were child-selected activities such as center time during AM, and free-play during PM. For the Fall AM coded videos for each child, we excluded from our analysis whole-group activities in which an adult addressed multiple children at once (e.g., circle time and read aloud by the teacher). These activities were identified by the first author, who watched each video in its entirety and timestamped whole-group activities for occurrence, which allowed their exclusion. The averaged employed recording time per focal child was 40.58 minutes in Fall AM (range = 35.61 to 46.38 min), 40.78 minutes in Fall PM (range = 27.38 to 57.34 min), 40.88 minutes in Winter AM (range = 30.80 to 58.38

min), and 47.10 minutes in Winter PM (range = 29.18 to 59.13 min). The average employed recording time for each focal child per day of observation (AM and PM) was 78.6 minutes in Fall (range = 65.23 to 99.92 min) and 91.18 minutes in Winter (range 65.59 to 112.93 min).

**Manual coding and transcription of CDS.** Reference (ground truth) representation of children's exposure to CDS from both adults and peers in the classroom is drawn from manual coding and transcription of Fall AM recordings used in a previous study. This coding and transcription occurred over several steps and was conducted by trained, reliable research staff who met a priori guidelines to deem them reliable coders and transcribers. Each coder was required to complete a conceptual training and multiple practice sessions where the trainees' coding and transcripts were compared against goal standards. Furthermore, for the coding of focal children's interactions, 11% of the videos, randomly selected, were routinely double-coded to monitor and prevent drift. After coding each video, discrepancies between coders were discussed until agreement was met. The coding and transcription of focal children's recordings were conducted separately.

Focal children's interaction with peers and adults were coded using the Datavyu software. To capture each interaction instance, representing the event of interest, the authors created a time-event coding system to record when the interaction started and when it finished. Peer interaction was defined as any instance in which a focal child physically or verbally engaged with another child, or group of children, including playing, learning or working together, and fighting. Adult interactions were operationalized as any instance in which a focal child engaged with an adult in the classroom, including comforting (verbally or physically) the focal child, disciplining, problem-solving, directing, and playing. Ten percent of the coded videos were selected for double coding, and intraclass correlations (ICCs) were calculated to examine the reliability. The absolute-agreement 2-way mixed-effects model ICC for the duration of each category was greater than 0.97.

Only the talk directed to the focal children (i.e., CDS) was verbatim transcribed using the Systematic Analysis of Language Transcripts (SALT) software. To be considered CDS, peers and teacher's talk should meet at least one of the following criteria: (a) peers' or adults' behaviors involved the focal child while they talked (e.g., through physical contact or gaze direction), (b) peers or adults posed questions directed to the focal child, contained the focal child's name, or were on the same topic as a focal child's preceding utterances and are not utterances addressed to another person, and (c) peers' and adults' utterances occurred up to 3 seconds after the focal child's utterance. Peers' and teachers' utterances that did not meet these criteria were not transcribed. Two well trained English-speaker research assistants completed the transcripts, and one author of this paper check their accuracy by comparing the transcript against the video word-by-word.

**Automatizing coding and transcription of CDS.** Given that the CDS transcription was not timestamped in the original dataset, we operationalize CDS as talk occurring *within* an interaction with another person, including peers and adults. Tables containing each child's interaction's onset and offset were transformed into time trajectories to serve as reference (ground truth) to the RI detectors in Matlab. In addition, the following measures were calculated using Matlab's text analytics toolbox [22] at the individual-child level: interaction frequency, median and standard deviation of duration, and total duration of interactions. For CDS, we computed the number of utterances (NU), total number of words (TNW), number of different words (NDW), as well as the composite measures mean length of utterance (MLU = TNW/NU) and type-token ratio (TTR = NDW/TNW).

**Multimedia analysis pipeline.** Fig 1 provides a schematic view of the training and validation pipeline of the CIDR system. The system comprises four basic components: (a) scene parsing and featurization module, (b) feature extraction module, (c) interaction detector, and

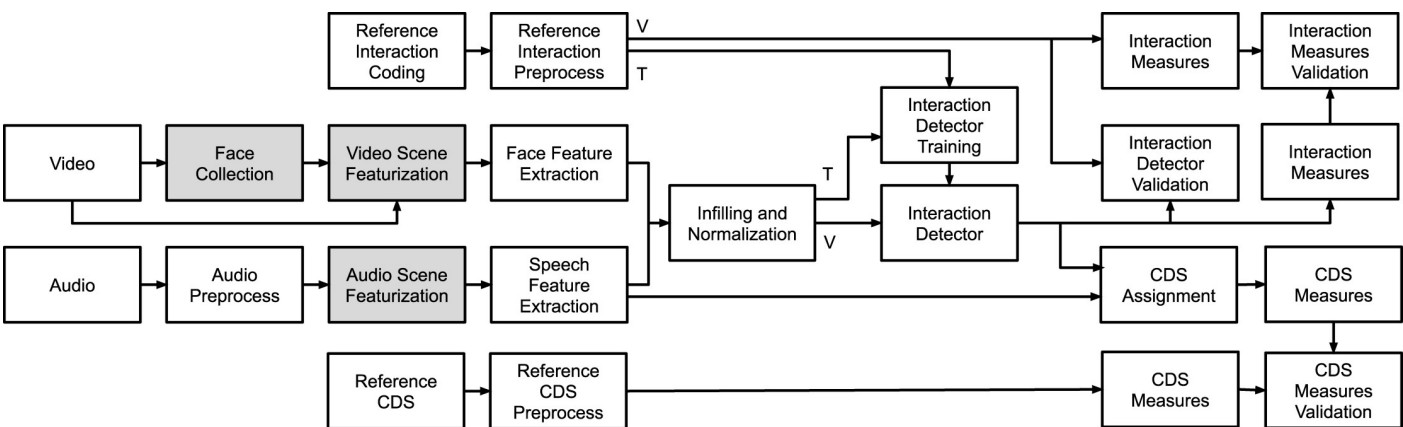

**Fig 1. Schematic view of the pipeline of the Classroom Interaction Detection and Recognition (CIDR) system.** Abbreviations correspond to child-directed speech (CDS), validation branch (V) and training branch (T). Boxes in light gray correspond to featurization modules deployed using Amazon Web Services (AWS).

(d) mappings to interaction and CDS measures. Boxes in light gray correspond to featurization modules, which were the only elements in the pipeline that are not open-source in this work, with the rest of the elements implemented as Matlab and Python code, available at http://github.com/hugonvilla/CIDR.

*Scene parsing and featurization module*. We opted to get high-level audio and video features using Amazon Web Services (AWS), specifically their computer vision service called *Amazon Rekognition* [23], and their automatic speech recognition (ASR) service called *Amazon Transcribe* [24]. Both services are offered in a pay-per-use format, which, at the time of this publication, is $0.10 per minute of video files, and $0.00004 per second of audio. While there exist open-source alternatives for audio and video featurization that could replace AWS (see references in [16, 25]), the rationale behind the AWS choice included ease of use, low cost, satisfactory performance results on preliminary analyses with our dataset, and data security measures provided via institutional access to the cloud computing resources. AWS deep neural network models are trained with Amazon's proprietary data, and their online tutorials enable their deployment by non-experts. *Rekognition*'s face detection and face recognition accuracy were independently evaluated in [26], while *Transcribe*'s ASR performance in conversational settings was assessed in [27], both with satisfactory results for our application. *Rekognition* face detection output was employed to study infant behavior in [28].

The audio-recordings were retrieved as 32 kHz mono signals, which were down-sampled to 16 kHz before the automatic transcription, following Transcribe guidelines. Amazon Transcribe returns transcription of speech segments, their onset and offset, and a speaker label for each segment (i.e., diarization). We noted that the speaker identification module returned different speaker labels for the same speaker captured at different distances from the focal child microphone, so we did not use that information. As an alternative, we tested an existing speaker recognition algorithm employed with audio from child-worn recorders in home settings. However, its performance was inadequate for our application, possibly due to the different nature of our testing dataset (preschool classroom, 3–5 year-old children). We designed an audiovisual labeling tool in Matlab to manually label each utterance detected by Amazon Transcribe as being uttered by an adult, the focal child, or a peer, irrespectively of the addressee. Our future work contemplates training existing automatic speaker recognition algorithms using the labeled data described in the previous sentence.

Face detection (where is a face?) and recognition (whose face is that?) were employed to parse the visual scene. The face detection module, *FaceDetect*, returns 30 facial landmarks and 15 facial features, including age range, head pose, and mouth open confidence for each detected face. The average temporal resolution for the face detection results was 500 ms. The facial recognition module, *FaceSearch*, was used only to remove non-teacher adults' data in the video scene (primarily research staff), since our reference (ground truth) only discriminates focal child interactions with teachers in the classroom and peers. To recognize the faces, we selected video frames containing children and teachers' faces following the guidelines in [23]. At least three images per child and teacher were employed to build a *Face Collection*, an Amazon Rekognition object that assigns a face ID to each face in the image. *FaceSearch* compares the faces in the video frames with the face ID to recognize faces. We also wrote a Matlab code to correspond each face ID with the participant ID for each child. AWS preserves facial landmarks, and not actual images, in the face collection. Besides the face ID, the *FaceSearch* module returns the dimensions of the face bounding box and the head pose for each face.

*Feature extraction module.* To harmonize the scene featurization response and extract the features to train the interaction detector, we wrote Matlab codes to process the *Amazon Transcribe* and *Amazon Rekognition* results. We selected the following signals as the audio features for the interaction detector: i) a Boolean variable corresponding to speech activity from the timestamps of the automatic transcription, ii) the average word confidence score for each segment, and iii) the perceived loudness of each segment, using the *acousticLoudness* function from Matlab's Audio Toolbox. The latter implements the norm ISO 532–1 to calculate loudness as perceived by persons with normal hearing [29], and was regarded as a proxy for acoustic proximity. We processed the transcript to separate language contractions in the same manner as the manual transcription (e.g., you're to you are). As an approximation to the utterance segmentation approach used for the manual transcription process, we segmented utterances using the punctuation marks returned by Amazon Transcribe as boundaries. From the video scene featurization, we aligned the outputs of *FaceDetect* and *FaceSearch* by matching timestamps and face bounding boxes. In this stage, we only allowed FaceSearch results with a similarity index above the threshold of 92%, set by iterative experiments, to avoid misidentified faces. Since the number of faces at any frame is variable, and most machine learning methods accept a fixed number of features as input, we selected the features of the *most representative* faces on each frame as inputs to the interaction detector. Based on iterative experiments, we defined the most representative face in terms of the faces' bounding box size, which is a proxy used for distance to the focal child [30]. The features from these faces are the bounding box size, yaw, roll, and pitch of the face, and the probabilities (confidence) for mouth open and eyes open.

*Preprocessing and interaction detector.* We preprocessed the output of the scene featurization modules prior to running the interaction detector. The output of the scene featurization modules might not capture the fast temporal dynamics occurring during an interaction, and our reliance on facial features made us prone to problems with face occlusion. To counteract these issues, we reprocessed the features via a high-temporal resolution infilling using moving averages, setting the sample period of our feature signals to be 250 ms. Outliers beyond the 99th percentile were removed using the clip method, and features were normalized before the RI detector training routine.

Given the distinct nature of adult and peer interactions in terms of frequency and duration as captured in the manual coding [21], we achieved higher accuracy when training separate classifiers for interactions with adults versus peers. Long short-term memory (LSTM) are artificial recurrent neural networks often employed in sequential data classification problems, such as voice activity detection, accelerometry, and video analysis [31]. We defined a deep-

learning interaction sequence classifier using Matlab's deep learning toolbox [32]. The classifier consisted in a sequence input layer, a bi-directional LSTM (BILSTM) layer with 200 hidden units, a softmax layer, and an output classification layer, trained using 30 epochs with all 12 facial features (6 for adults, 6 for peers) and the audio features for both the adult interaction detector and the peer interaction detector. Thus, both detectors have the same feature set, and we will refer to them as full-feature BILSTM. We also trained the same detectors using only the face bounding box area (face size) and speech activity, which we refer to as reduced-feature BILSTM. The purpose of testing the reduced-feature BILSTM is to mimic less invasive data collection tools like indoor positioning systems. In a post-processing module, the predicted RI trajectories are smoothed by merging or deleting predicted interactions with duration of less than one second, following results from [21].

*Mappings to interaction and CDS measures.* A final step was the diarization of CDS, which involved assigning each predicted utterance to either peer, adult, or non-CDS based on the predicted interaction trajectories. If there is an intersection between the time segments of an utterance and an interaction with adults, then the utterance is considered labeled as adult-CDS. Similarly, if the intersection is with an interaction with peers, the utterance is labeled as peer-CDS. If both a peer-CDS and adult-CDS was predicted simultaneously, the utterance was assigned to an adult. Lastly, if there is no intersection between utterance and interaction time segments, the utterance is labeled as non-CDS. During preliminary tests, the word-by-word accuracy achieved by Amazon Transcribe with our hardware setting in a noisy acoustic environment was not suitable to compute grammatical complexity measures, with word error rates (WER) above 0.6. However, we obtained acceptable estimates for the quantity of talk, including number of utterances, number of words, and number of different words.

We calculated interaction measures using the time trajectories of the reference and predicted interactions. After computing the onset and offset of each interaction, we calculated the frequency, median duration, and standard deviation of the duration of interactions. We used Matlab's Text Analytics toolbox to tokenize the segmented automatic transcription from Amazon Transcribe, and computed the number of utterances, mean length of utterance, and total number of different words following the transcription manual [21].

## Results

### Can facial features and speech activity predict interactions in a preschool classroom?

This study used a leave-one-subject-out (LOSO) cross-validation method to validate the CIDR system and determine whether speech activity and high-level facial features, operationalized in the full-feature BILSTM, can predict interactions more accurately than employing speech activity and face size, operationalized in both the reduced-feature BILSTM and the thresholding methods. The LOSO cross-validation method consists of training the model with data from all but one subject and subsequently testing it with data corresponding to the held-out subject.

All Fall AM recordings corresponding to 13 children were employed in the cross-validation procedure. The performance of the unsupervised reduced features BILSTM and full BILSTM detectors were compared with the reference RI obtained from manual coding. To assess the performance of the interaction detectors, we measured accuracy, precision, recall, and the F1 score. Accuracy is computed as the ratio between the number of 250-ms time samples correctly labeled as interaction or not-interaction and the total number of time samples in the sequences. Precision can be interpreted as the probability that a randomly selected 250-ms time sample labeled as interaction by the detector is indeed a true interaction, whereas Recall

Table 1. Leave-one-subject-out (LOSO) performance results for the detector of focal child interactions with adults and peers.

| Model | Adult Interaction Detector | | | | Peer Interaction Detector | | | |
|---|---|---|---|---|---|---|---|---|
| | Acc | Prec | Rec | F1 | Acc | Prec | Rec | F1 |
| Reduced features | 80.8 | 90.0 | 85.4 | 87.6 | 86.1 | 98.4 | 87.0 | 92.3 |
| Full features | 81.1 | 92.3 | 84.2 | 88.1 | 85.2 | 96.5 | 87.4 | 91.7 |

Note: Reduced features model correspond to bi-directional long-short term memory (BILSTM) network trained with face size and speech activity data, while the full feature model is a BILSTM network trained with the full set of facial features and speech activity. The performance metrics considered are accuracy (Acc), precision (Prec), recall (Rec), and F1 score (F1).

is the probability that the interaction detector identifies a true interaction time sample. The F1 score combines precision and recall via the harmonic mean.

Table 1 shows LOSO validation results for adult and peer interaction detectors. For the adult interaction detector, results show that the supervised full BILSTM classifier, trained with full facial features, achieves higher accuracy and F1 score (81.1 and 88.1, respectively) compared to the reduced feature detector (80.8 and 87.6, respectively). For the peer interaction detectors, the reduced features detector obtains higher accuracy and F1 score (86.1 and 92.3, respectively) than the detector trained with full features (85.1 and 91.7, respectively). While both detectors' performance is adequate for our application, we selected the full feature detector for further analysis, as it performed better on a separate hold-out validation run.

## Can facial features and ASR detect and measure CDS in a preschool classroom?

To measure the CIDR's performance on CDS measures, we employed five variables for focal children's interaction with adults and peers. The five variables included total number of utterances (TNU), total number of words (TNW), number of different words (NDW), mean length of utterance (MLU), and type-token ratio (TTR). The MLU is the ratio between the total number of words and the total number of utterances, while the TTR is the quotient between the number of different words and the total number of words. Table 2 shows LOSO performance results for adult CDS and peer CDS measures, and measures of focal child speech directed to adults and peers detected during the predicted interactions obtained using the BILSTM classifiers. The raw measures correspond to those calculated using the raw output of the ASR service. All measures were normed by 10 minutes to facilitate comparison amongst children with

Table 2. Leave-one-subject-out (LOSO) cross-validation results for the CDS recognition system.

| Speaker | | TNU | | TNW | | NDW | | MLU | | TTR | |
|---|---|---|---|---|---|---|---|---|---|---|---|
| | Adaptation | mARE | r | mARE | r | mARE | r | mARE | r | mARE | r |
| **Adult** | Raw | 39.0 | 0.75 | 51.3 | 0.79 | 24.3 | 0.84 | 19.7 | 0.32 | 47.2 | 0.88 |
| | Adapted | 12.9 | 0.69 | 34.9 | 0.73 | 9.7 | 0.80 | 7.0 | 0.18 | 12.7 | 0.85 |
| **Peer** | Raw | 92.9 | 0.46 | 96.1 | 0.50 | 93.3 | 0.49 | 47.6 | 0.44 | 88.7 | 0.52 |
| | Adapted | 40.6 | 0.36 | 40.3 | 0.38 | 20.8 | 0.37 | 6.9 | 0.34 | 25.5 | 0.42 |
| **Focal** | Raw | 76.4 | 0.44 | 88.7 | 0.44 | 75.0 | 0.54 | 37.2 | -0.11 | 100.0 | -0.56 |
| | Adapted | 40.9 | 0.38 | 47.8 | 0.37 | 27.6 | 0.48 | 24.0 | -0.58 | 23.5 | 0.51 |

Note: Language measures are total number of utterances (TNU), total number of words (TNW), number of different words (NDW), mean length of utterance (MLU), and type-token ratio (TTR). The performance metrics depicted are: median absolute relative error (mARE) and linear correlation (r). Raw refers to non-adapted measures calculated using the raw output of the automatic speech recognition service.

different recording times. The data presented in Table 2 shows the median values for the absolute relative error (ARE) for these variables for each speaker. For any of the above five measures, denoted with $X$, the corresponding ARE for focal child $j$, represented by $E_{X,j} \geq 0$, is computed by measuring the deviation between the values obtained by using the manual (reference) transcription $X_{r,j}$ and predicted transcripts $X_{p,j}$ by

$$E_{X,j} = \frac{|X_{r,j} - X_{p,j}|}{X_{r,j}} * 100 \qquad (1)$$

with values close to zero representing less deviation from the reference values. Table 2 also presents linear correlations between the reference and predicted measures, denoted by the vectors $X_r = [X_{r,1}, \ldots, X_{r,n}]$ and $X_p = [X_{p,1}, \ldots, X_{p,n}]$.

It can be seen that the median absolute relative errors (mARE) for the measures corresponding to the adult speakers are lower and linear correlations (R) than peer and child measures, which is not surprising, considering the lack of performance on recognizing child speech by current ASR services [33]. For our sample, the correlation between age of the focal child in months (between 35 and 58 months) and the mARE of TNW for the focal child was of r = -0.44, indicating a tendency for the ASR performance to improve with age. For both adult and children, the raw ASR output underestimates the TNU and the TNW, due, in part, to the noisy acoustic environment. To take advantage of measures with high correlations (r) in Table 2, we adapted (calibrated) the raw results using ordinary least products (OLP) regression with measures obtained with manual transcriptions, following [34]. The OLP, or Type II regressions were done as part of the LOSO validation loop. A similar adaptation, based on ordinary least squares, was employed in [35] to estimate word count on daylong recordings. The LOSO performance using the OLP adaptation is also shown in Table 2, showing lower mARE compared to raw measures.

Although there are no commonly accepted conventions for interpreting Table 2 results in the context of CDS in preschool classroom, we can interpret them relative to other tools that allow for automatic word count estimation in home and laboratory settings with early childhood population. For instance, one study reported that when analyzing audio recordings containing adult and child speech, the median ARE valued for TNW estimation ranges between 10% and 80% for an open-source estimator, and between 28% and 60% when using LENA hardware and software [35].

Another recent study reviewing LENA performance [17] showed that the mean correlation between manually transcribed TNW and LENA adult word counts (adult TNW) was R = 0.79 with a mean relative error (MRE) of 13.76%, whereas the same metrics, when applied to child vocalizations (focal child TNW), was R = 0.77 and MRE of -40.48%. The computation of the relative error rate is done discarding the absolute value of the differences in Eq 1. For our adapted results, we obtained MRE for adult TNW of -1.8%, and -16.1% for focal child TNW. Taken together, these results indicate that our CDS detector and measurement system is adequate for adult TNW, while also providing estimates of NDW and TTR with similar performance. While our system achieves a MRE for focal child TNW closer to zero, as compared to average LENA results, the correlation with reference values remains lower, so more work is needed to accept these results as adequate. The low performance values on both peer and focal child's MLU and TTR's composite measures signify the challenge to automatically obtain these measures with the existing automatic speech recognition services.

Finally, we report LOSO-validation performance values for total duration of interactions (TDI) with adults and peers. For TDI with adults, the raw MRE is 23.6% with correlation r = 0.67, while for TDI with peers, the raw MRE is 62.7% and r = 0.47.

### Are there differences in children's CDS exposure within a classroom day (morning to afternoon) and over time (fall to winter)?

To apply the CIDR system to data sources that have not been manually coded or transcribed, we applied the system as depicted in Fig 1 to three additional data sources collected on the same set of children, representing Fall PM, Winter AM, and Winter PM video-recordings. To do so, we used data for children who had recordings in all four observed sessions (n = 9). We examine children's interactions and CDS exposure vertically within a classroom day (morning to afternoon) and horizontally over time (fall to winter) to explore the stability of children's interactions and CDS across time.

We conducted repeated measures analysis of variance (rANOVA) with time as the between-subjects factor for each measure, revealing no significant differences for most CDS measures during interactions with adults between morning and afternoon and fall to winter. However, measures concerning focal child speech were different between morning and afternoon for TNW (p = 0.04) and NDW (p = 0.03). While the validity of these tests is unclear due to the small sample size and modest correlation between predicted and reference measures for focal child speech, the boxplots in Figs 2 to 4 indicate trends in the classroom's linguistic environment. Fig 2 shows results for adult TNW, Fig 3 for peer TNW, and Fig 4 for focal child TNW during the four observation periods. As an aid to interpret the boxplots, the middle bar represents the median TNW, whereas the box edges correspond to the lower (0.25 quantile) and upper (0.75 quantile) quartiles. The whiskers (vertical lines) connect to the non-outlier maximum and minimum values; outliers are values that are more than 1.5 IQR (interquartile range) away from the box edges. It can be seen that median adult TNW is higher during the morning sessions consisting largely of a structured classroom task (center time), while both median peer and focal child TNW is higher during unstructured afternoon sessions.

## Discussion

This study was designed to develop, validate, and test an automatized system for the detection and recognition of interactions and CDS in a preschool classroom environment. The study builds on decades of research showing the importance of CDS as instrumentally important in facilitating young children's language development [2, 6, 7, 9, 36] and technological efforts to improve collection and analysis of CDS for a variety of scientific needs [10, 15, 35, 37, 38]. Here, we summarize and expand upon several primary findings in this report.

We obtained first-person video recordings using head-mounted cameras worn by children in a preschool classroom during Fall and Winter of the school year. The average employed recording time per child per day observation was 81.62 minutes, across child-selected activities (center time and free play). Decoding these classroom tasks is relevant to explain the higher heterogeneity in interactions and CDS during child-selected activities, compared to large-group activities [21]. While studies analyzing talk in home environments employ audio recordings with longer duration for ecological validity and reliability of speech processing tools [16, 39], our recording times are consistent with observational studies analyzing interactions and CDS in preschool classrooms [40–42].

First, teachers' and peers' facial features and speech activity, obtained using computer vision and automatic speech recognition technology from first-person video recordings accurately predicted instances of interactions in a preschool classroom with the focal children. We used manually coded interactions employed in [21] and developed two bi-directional long-short-term memory (BILSTM) classifiers: one with high-level facial features and speech activity as inputs, and the other with only proximity and speech activity as inputs, with face size as a proxy for proximity. The latter was trained to resemble non-invasive methods to detect

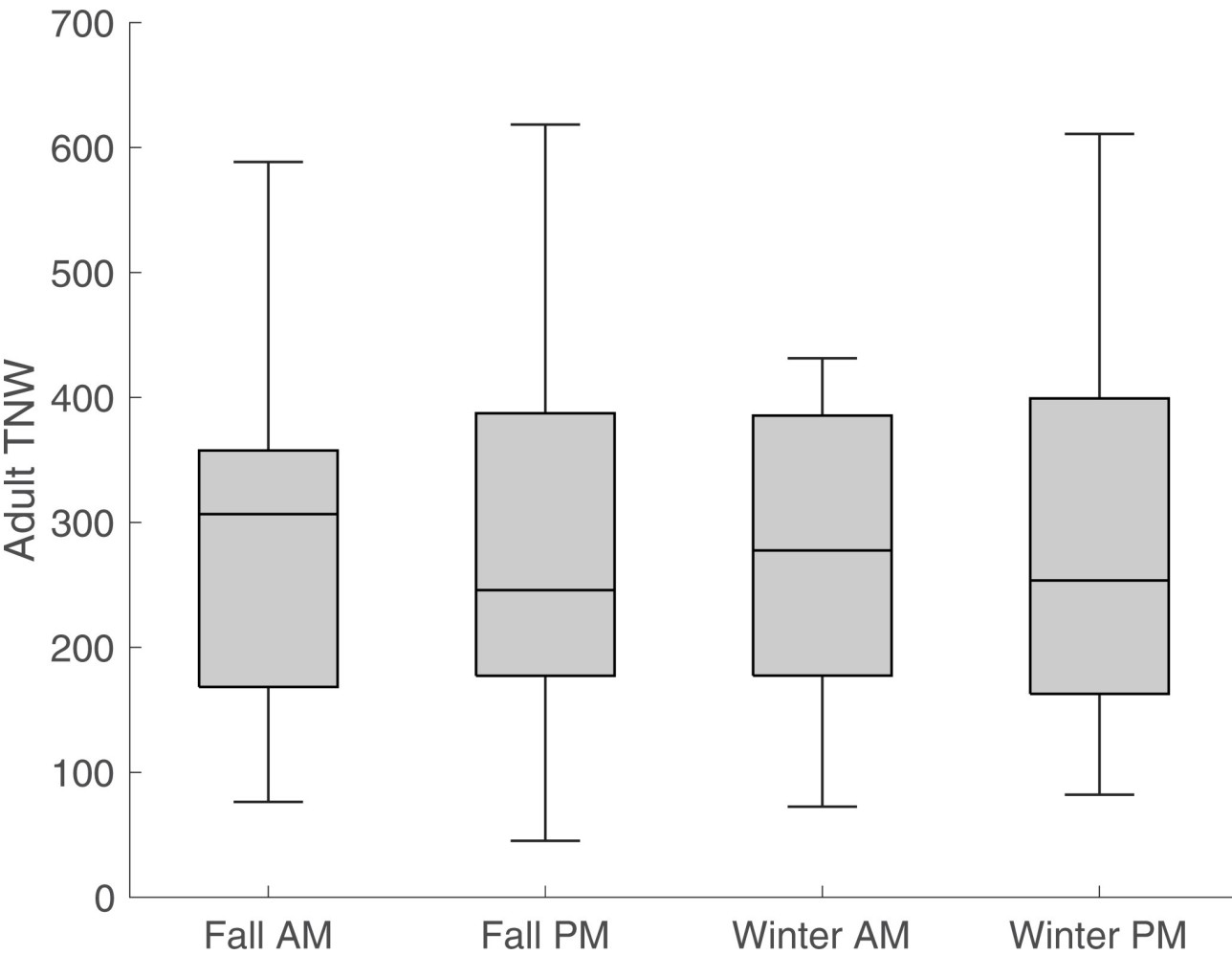

**Fig 2. Total number of words (TNW) by adult speakers during interactions with focal child.** Box edges represent lower and upper quartile, while middle line corresponds to the median. Whiskers depict minimum and maximum non-outlier values. Measures are normed at 10 minutes.

interactions via indoor position sensing systems [38, 43]. For detecting interactions with adults, the BILSTM classifier using high-level facial features outperformed the reduced features detector and detected instances of interactions with 81.1% accuracy. Interactions with peers were sparse and short, and the BILSTM classifier trained with proximity features achieved higher accuracy (86.1%). Optimizing the BILSTM classifiers' architecture could improve their performance and provide a clear answer on whether invasive but data-rich sensing systems, like first-person video recordings, are needed to detect interactions more accurately than less-invasive proximity sensors like RFID.

Second, we found that, in combination with the BILSTM detector, children's exposure to CDS within a preschool classroom can be detected and measured using automatic speech recognition (ASR) services. While the word-by-word accuracy of the ASR service was low, we obtained acceptable absolute error rates (ARE) on five measures of CDS during interactions with adults and peers, including total number of words (TNW) and number of different words (NDW). The mARE percentages for adult TNW are within the range of what has been observed in other automatic coding of the linguistic and social environment in early childhood

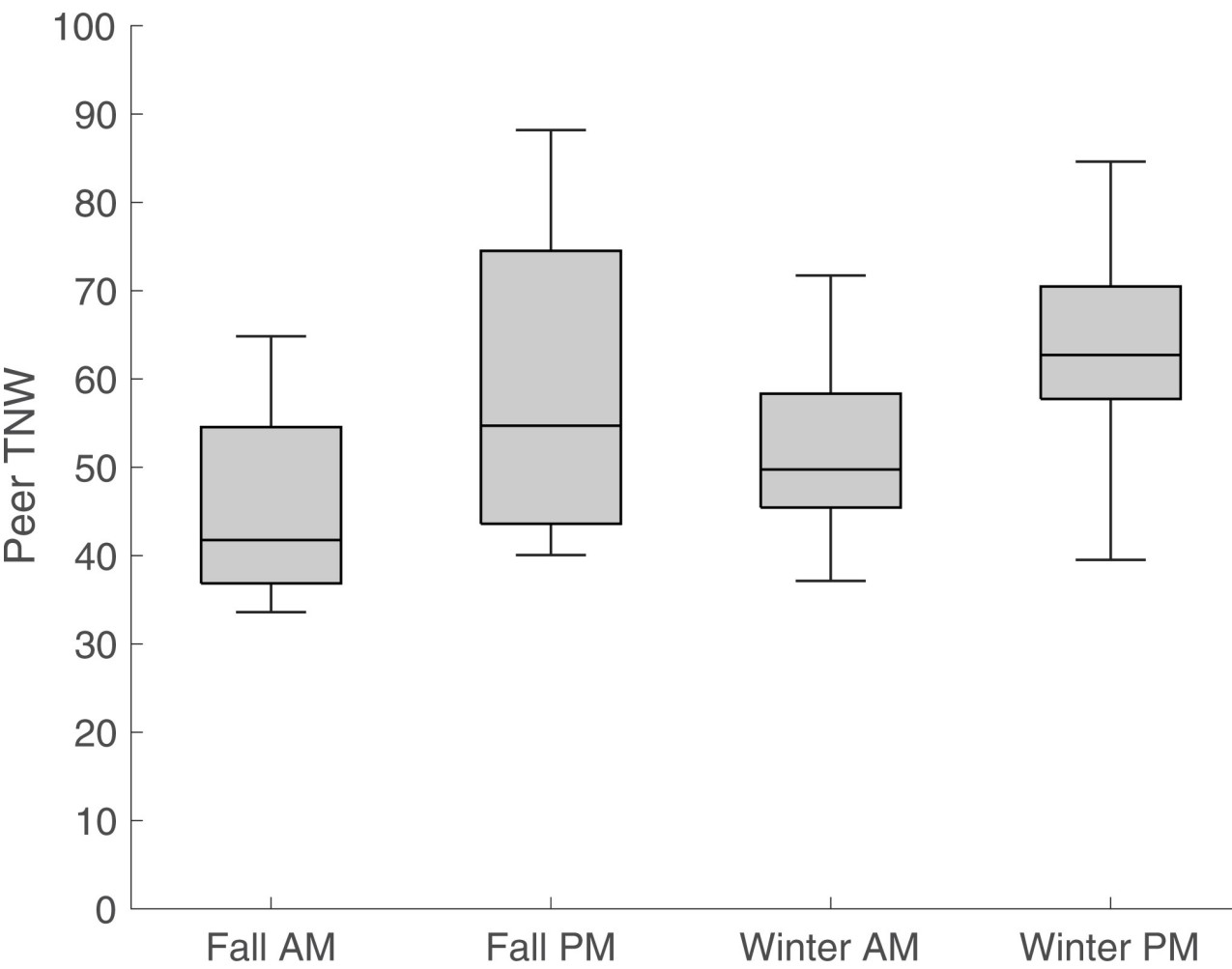

**Fig 3. Total number of words (TNW) by uttered by peers during interactions with focal child.** Box edges represent lower and upper quartile, while middle line corresponds to the median. Whiskers depict minimum and maximum non-outlier values. Measures are normed at 10 minutes.

settings like LENA [17, 35]. While our system achieves a mean RER for focal child TNW closer to zero, the correlation with reference values was modest compared to average LENA results. Importantly, unlike our audiovisual system, audio-only sensing systems like LENA have limitations in their abilities to accurately discriminate CDS from overheard talk [44]. These results are promising since there are several avenues to improve the results, particularly in terms of detecting and recognizing child speech [33].

Third, we used automated coding of children's interactions and exposure to CDS to examine children's interactions across one day and over time longitudinally. Analysis of these data did not reveal significant differences in five CDS variables. However, Fig 2 depicts higher volume of adult speech, measured by total number of words (TNW), during structured morning classroom task (center time), while higher exposure (TNW) to peer CDS, accompanied by higher focal child TNW during unstructured afternoon sessions (free play) are observed in Figs 3 and 4. Given the increased interest in peer-mediated interventions in preschool classrooms [45, 46], these results indicate the importance of identifying features of interactions with peers to optimize the design of such interventions.

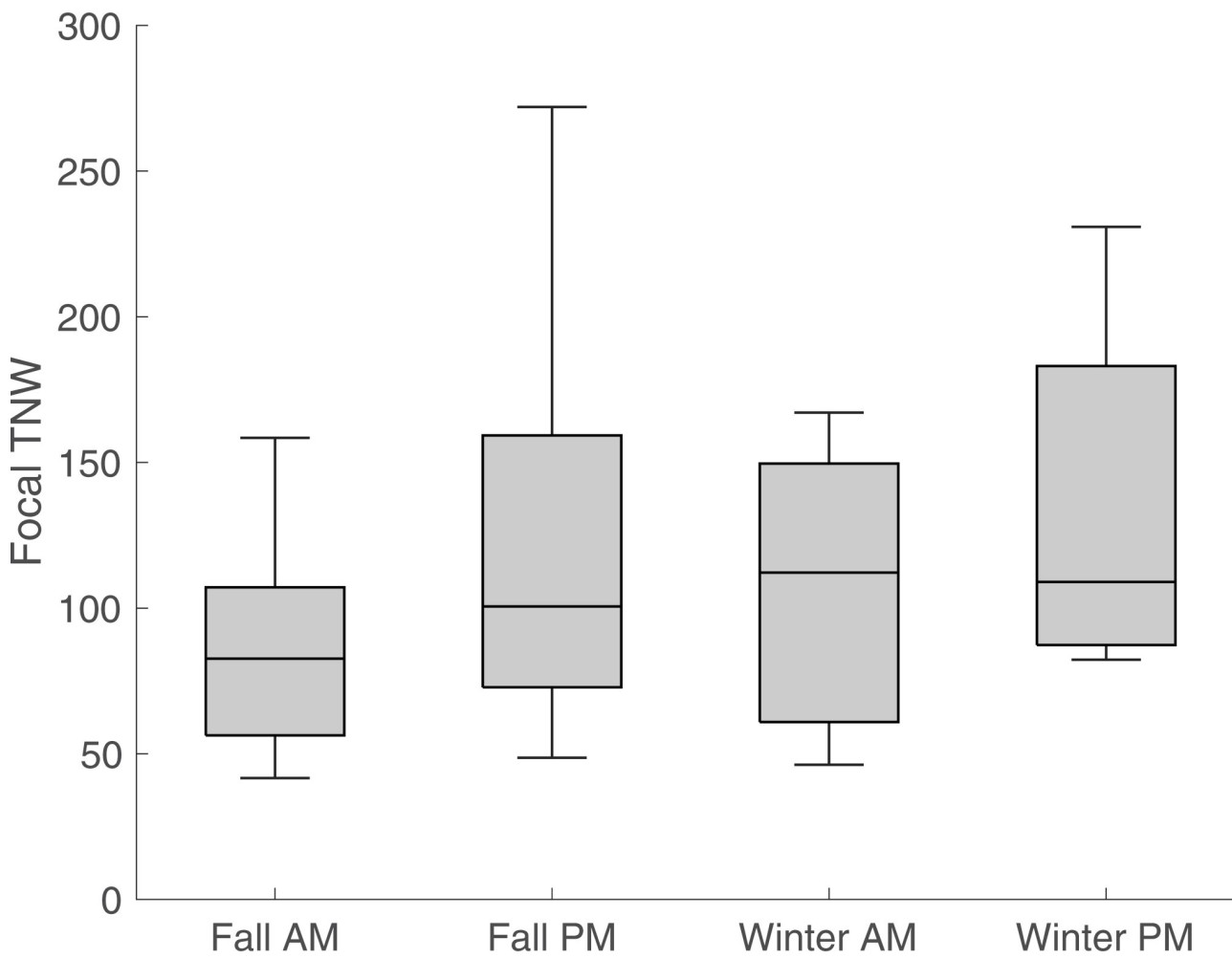

**Fig 4. Total number of words (TNW) by focal child during interactions with adults and peers.** Box edges represent lower and upper quartile, while middle line corresponds to the median. Whiskers depict minimum and maximum non-outlier values. Measures are normed at 10 minutes.

### Limitations and future research

Besides of our small sample size, limiting our ability to perform inferential analysis, most of the conceptual and technical limitations in this report refer to the audio and video signal processing. Since we did not have timestamped CDS transcriptions, the fact that we detected CDS *within* manually labeled and timestamped interactions could have introduced errors in the reference (ground truth) signal we employed to train our classifiers. We plan to continue improving our model with timestamped CDS transcripts. The video scene featurization relied on facial detection and recognition, so it was prone to problems with face occlusion during interactions. Future improvements could take advantage of body pose detection methods applied to educational contexts [25], as well as people pathing to improve people detection during interactions. The audio hardware used in this work was also a limitation for automatic decoding. Our preliminary testing suggests that using small lavalier microphones increases ASR performance, as measured by word error rate (WER) and diarization error rate (DER). Employing more ergonomic cameras with a wider field of view can also improve results by reducing occlusion and increasing recording times to include preschool tasks that were not feasible to record with our current setting, such as recess, meals, and naptime. Finally, we

expect the performance of the CIDR system to improve by increasing the training data set with more labeled data, or data augmentation methods, as well as alternative adaptation methods, including focal child age as an independent variable, given the negative correlation between age and mARE reported in this work. We also expect that by employing publicly available corpora in the training data set, we could extend CIDR's applicability to other settings, such as the home environment.

Future research can take advantage of CIDR's facial recognition module to identify which adults and peer the focal child interacted with. This would enable the study of the topology of preschool classrooms' social networks [47], which is an important and emerging area of interest. Furthermore, the facial and body pose detection could be employed to study features of "direct talk" that support positive change in language growth, including joined attention, emotionality during the interactions obtained via facial and speech emotion detection, and temporal adaptations of language between interacting partners. We also plan to train diarization routines using the labeled automatic transcriptions built for this work, to automatically discriminate between speech from adults, peers, and focal child, as in [48].

## Conclusion

In this study, we employed machine learning to develop, validate, and test an automatized system for detection of recognition of interactions and child-directed speech (CDS) in a preschool classroom environment. Our audiovisual sensing system based on computer vision and automatic speech recognition provided facial features and speech activity as inputs to a deep-learning classifier that detected instances of social interactions, allowing us to accurately estimate CDS measures. We tested our system by measuring five CDS variables during four observation windows, revealing the role of classroom tasks on quantity and quality of CDS exposure during interactions with adults and peers. We identified research directions to improve the system's performance and directions to apply the system in longitudinal studies in language and social-emotional development.

## Author Contributions

**Conceptualization:** Hugo Gonzalez Villasanti, Laura M. Justice.

**Data curation:** Leidy Johana Chaparro-Moreno.

**Formal analysis:** Hugo Gonzalez Villasanti, Laura M. Justice.

**Methodology:** Hugo Gonzalez Villasanti, Laura M. Justice, Tzu-Jung Lin, Kelly Purtell.

**Software:** Hugo Gonzalez Villasanti.

**Writing – original draft:** Hugo Gonzalez Villasanti, Laura M. Justice, Leidy Johana Chaparro-Moreno, Tzu-Jung Lin, Kelly Purtell.

**Writing – review & editing:** Hugo Gonzalez Villasanti, Laura M. Justice, Leidy Johana Chaparro-Moreno, Tzu-Jung Lin, Kelly Purtell.

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
