## [Decision Letter · Decision Letter 0]

24 Aug 2020

PONE-D-20-15910

Automatized Analysis of Children’s Exposure to Child-Directed Speech in Preschool Settings: Validation and Application

PLOS ONE

Dear Dr. Gonzalez Villasanti,

Thank you for submitting your manuscript to PLOS ONE. I now have reviews from two experts in the field. While both reviewers felt your submission could make an excellent contribution to the field, one significant concern was identified. Reviewer #2 notes that the data recorded from individual children is limited, raising concern regarding its adequacy. My own review of your manuscript prompted the same concern. Typically this problem would be sufficient to warrant a more negative decision, but I think you need to be given a chance to demonstrate that the amount of data was sufficient to perform the analyses that you did. If you are able to provide evidence that your data are reliable then we can reconsider it for publication.  Therefore, we invite you to submit a revised version of the manuscript that addresses the points raised during the review process.

We look forward to receiving your revised manuscript.

Kind regards,

Susan Nittrouer, Ph.D.

Academic Editor

PLOS ONE

Journal Requirements:

2. You indicated that you had ethical approval for your study.

In your Methods section, please ensure you have also stated whether you obtained consent from parents or guardians of the minors included in the study or whether the research ethics committee or IRB specifically waived the need for their consent.

Reviewers' comments:

Reviewer's Responses to Questions

**Comments to the Author**

1. Is the manuscript technically sound, and do the data support the conclusions?

Reviewer #1: Yes

Reviewer #2: Partly

2. Has the statistical analysis been performed appropriately and rigorously? 

Reviewer #1: Yes

Reviewer #2: Yes

3. Have the authors made all data underlying the findings in their manuscript fully available?

Reviewer #1: Yes

Reviewer #2: Yes

4. Is the manuscript presented in an intelligible fashion and written in standard English?

Reviewer #1: Yes

Reviewer #2: Yes

5. Review Comments to the Author

Reviewer #1: This article presents results for a system to detect child directed speech through audio and visual analysis. The authors make use of existing open-source and cloud-based services for feature extraction and transcription. Additionally, the authors agree to make all code and data available.

While I have not worked directly in this area of research, the article is clearly written, well rationalized, and very accessible.

I have no edits to suggest.

Reviewer #2: General Comments:

This manuscript focuses on developing an automatic system to recognize child-adult and child-child interactions and CDS (child directed speech) in the preschool classroom environment. In this study, the primary subject (child) is referred to as the focal child and other children as referred to as peers. Research based on CDS has shown prior evidence that CDS might be an important factor to study the development of pre-school children’s language. Although professional collection, recording, and evaluation of CDS is a time-intensive process, technological advances based on machine learning models might aid the transcription of the spoken language, and hereby reducing the time and effort in studying CDS. However, in this study, apart from using the spoken language content the authors have also leveraged visual feed from cameras mounted on the subjects’ head.

please see comments in reviewer report.

6. PLOS authors have the option to publish the peer review history of their article (what does this mean?). If published, this will include your full peer review and any attached files.

Reviewer #1: No

Reviewer #2: No

---

## [Author Response · Author response to Decision Letter 0]

10 Sep 2020

Dear Editor, 

Thank you for coordinating the review of our paper titled ‘Automatized Analysis of Children’s Exposure to Child-Directed Speech in Preschool Settings: Validation and Application.’ The reviewers' comments were very beneficial, for they made us aware that some major points we were trying to convey remained unclear. We have addressed the comments (“C”) in the manuscript and provide the answers in this letter (“A”). Thank you for your consideration of this work.

Sincerely,

Hugo Gonzalez Villasanti, Laura M. Justice, Johana Chaparro Moreno, Tzu-Jung Lin, Kelly Purtell

Reviewer 2

Comment 1: The amount of data recorded was very limited. In the original Hart and Risley study which was published as “Meaningful Differences” – they also had short 1hr recordings. To be effective here, much longer recordings which are in naturalistic context are needed. The proof-of-concept idea could be assessed with 1hr recordings, but from my experience with LENA analysis and speech recognition/analysis for diarization (who spoke what when and how), 3-8hr recordings would provide better overall measures that are more consistent long-term.

A: In the revised manuscript, we expanded the discussion about this topic in lines 533-541 and 601-602, which we restate here. We detailed that the average employed recording time per child per day observation was 81.62 minutes, across child-selected activities (center time and free play). Decoding these classroom tasks is relevant to explain the higher heterogeneity in interactions and CDS during child-selected activities, compared to large-group activities (21). While studies analyzing talk in home environments employ audio recordings with longer duration for ecological validity and reliability of speech processing tools (39, 40), our recording times are consistent with observational studies analyzing interactions and CDS in preschool classrooms (41-43). We also noted that technological advances towards small, long-battery duration cameras would make it feasible to expand our analysis to include preschool tasks that were not feasible to record with our current setting, such as recess, meals, and naptime.

Comment 2: Line 73-25: yes, the LENA analysis software provides measures of Adult-word-count (AWC), child vocalizations (CV), and conversational turns (CT). The study ref’s [10] was done by LENA foundation and they did benchmark on 1hr blocks but that was to assess the changes over the day. The original Hart & Risley study was more focused on the daily word count exposure for the child. Betty Hart’s work emphasized the range of word exposure was important, and at least studies from LENA foundation folks showed (and in discussions with Todd Risley) was that the 1hr recorded analog cassette tapes in Hart’s work over predicted what might take place in actual daily language rich environments. So – _the question here is that these measures are all ESTIMATED from the audio stream. It would be helpful to bring this point out better here – _since the text here suggests that the LENA software “extracts” _this information vs. estimates it.

A: In the revised manuscript, we noted in lines 73-75 that the measures are estimated by the LENA software from audio signals. Thank you for pointing this out.

Comment 3: So, the publication [17] does point to some issues on accuracy of the LENA analysis software. It is again important to understand how that system was developed and intended to be used. LENA was set up for 1 adult and 1 child alone in a room having conversation engagement. When speech is captured by LENA recorder in diverse spaces, the analysis software was not intended to be used for accurate analysis. There have been a number of studies on robustness of speech technology in naturalistic settings and how performance degrades in mismatch conditions (i.e., differences between training and testing conditions for audio streams when automatic acoustic/language models are trained). This study gives a number of examples where research laboratories thought they were characterizing a speech related concept, but instead the technology they used was not intended for use in that context. I would agree that accuracy may not be consistent if the environmental context changes, but the text here should emphasize that the user therefore should use the technology only in the context it was intended, and if it’s different – note that the user (not the software/LENA) is applying this tool/solution in a scenario which they should expect increased error. As a side note, LENA technology/software has been used in many countries outside of the USA, and applied to languages other than English, however all the software tools were developed using English data, and therefore it cannot be expected to work in that context.

A: In the revised manuscript, we noted in lines 111-113 that studies reviewed in (17) utilize LENA in settings that do not match LENA’s training data, recorded on English-speaking home environments of children ages 1 to 42 months old (18).

Comment 4: so, it was surprising that the study used online ASR transcription support for the task here. The audio data is spontaneous and involves adult-child interaction, so it’s not expected that ASR performance would be very good. These online services are intended to be used more for directed adult speech for maybe lectures, or telephony scenarios. The level of coarticulation seen in naturalistic data streams like this would not be a good choice. Also – _it’s not clear that the corresponding Lexicon and Language Model for the ASR engine (Amazon of course has extensive LM and Lexicon advancements), but the specifics here for adult-child interaction for me would be very different in terms of the needed performance. To benchmark the ASR tool – _I might suggest having the adults actually read prepared text passages where ground truth is known, and compare the ASR performance for this single speaker, clean/noise-free audio ASR output with what might have taken place in the 1hr recordings

A: Regarding benchmark tests, in the discussion section of the original manuscript, we reported the word-error-rate of preliminary tests of the ASR on 2-min segments of classroom audio. For clarity, in the revised manuscript, we moved that discussion to the methods section between lines 359-363, including a comment about obtaining acceptable estimates for the quantity of talk, including number of utterances, number of words, and number of different words. We hypothesize that Amazon’s ASR (possibly end-to-end) service acts like a voice activity detection and word-count estimation on our noisy audio. Also, in lines 272-273, we referenced study (27) that evaluated the performance of Amazon Transcribe in conversational scenarios.

Regarding the ASR service's performance for estimation of CDS measures in our dataset, in our revised manuscript, we reworded the results section around Table 2, between lines 445 and 453, to improve clarity. We note that while the word count is under-estimated by the ASR service, we improved adult CDS estimation by exploiting the high correlation between some reference (truth) and estimated (predicted) measures. We adapted the latter using ordinary least products regression (OLP), as recommended in (34), with a similar application for word count estimation on home environments in (35).

Comment 5: A major limitation of this study, as reported by the authors (as well), is the automatic processing of the audio and video signals. Training the neural networks with features extracted from children or peers in this experiment, might improve the accuracy of child speech recognition models as compared to using the AWS neural networks. Being able to separate out the (i) performance of the tool, vs. (ii) the viability of the knowledge extracted in reflecting what is happening in ground truth context is important. It is in this area where there needs to be more work to move this manuscript forward. I would suggest collecting some probe data with the same camera/audio capture setup, but in controlled setting with single adult speakers. Have them produce speech by (i) reading prepared text, and (ii) producing spontaneous speech. With video capture, the process could also simulate interaction by having a picture of a general child in a screen when the read or spontaneous speech should be directed. In this control context, it would allow the authors to benchmark the viability of the technology without the child in a controlled context. This would allow for a more effective comparison when used in natural scenario. I feel the existing analysis, while it could have been done on larger data streams, has value and I would not require they have to expand that effort.

A: Regarding the viability of the employed tools, in the lines 271-274 of the revised manuscript, we reference independent studies assessing the performance of Amazon Rekognition and Amazon Transcribe, as well as an application of Amazon Rekognition on an early childhood setting. 

Editor

Comment 1: Reviewer #2 notes that the data recorded from individual children is limited, raising concern regarding its adequacy. My own review of your manuscript prompted the same concern. Typically this problem would be sufficient to warrant a more negative decision, but I think you need to be given a chance to demonstrate that the amount of data was sufficient to perform the analyses that you did. If you are able to provide evidence that your data are reliable then we can reconsider it for publication.

A: Thank you for raising this point. Our recording times are consistent with observational studies analyzing interactions and CDS in preschool classrooms (41-43), consequently within this particular context the data recorded for individual children is consistent with current research. We also noted that technological advances towards small, long-battery duration cameras would make it feasible to expand our analysis to include preschool tasks that were not feasible to record with our current setting, such as recess, meals, and naptime.

Editorial Office

Comment 1: You indicated that you had ethical approval for your study. In your Methods section, please ensure you have also stated whether you obtained consent from parents or guardians of the minors included in the study or whether the research ethics committee or IRB specifically waived the need for their consent.

A: Thank you for raising this point. In the revised manuscript, in the line 140 we indicated that our study was approved by The Ohio State University IRB. We obtained written consent for three teachers and primary caregivers of 19 children, as detailed in lines 148-155.

Note:

We want to note that we omitted to report a late update to the algorithm in the original manuscript. This update, detailed in lines 308-312 of the revised manuscript, was already incorporated in the algorithm that produced the original manuscript results, so no changes have been made to the results section. We apologize for the omission.

---

## [Editor Report · Decision Letter 1]

4 Nov 2020

Automatized Analysis of Children’s Exposure to Child-Directed Speech in Preschool Settings: Validation and Application

PONE-D-20-15910R1

Dear Dr. Gonzalez Villasanti,

Thank you for submitting your revised manuscript to PLOSONE. We’re pleased to inform you that your manuscript has been judged scientifically suitable for publication and will be formally accepted for publication once it meets all outstanding technical requirements. We appreciate your patience with the delay.

Kind regards,

Susan Nittrouer, Ph.D.

Academic Editor

PLOS ONE

---

## [Editor Report · Acceptance letter]

14 Nov 2020

PONE-D-20-15910R1 

Automatized Analysis of Children’s Exposure to Child-Directed Speech in Preschool Settings: Validation and Application 

Dear Dr. Gonzalez Villasanti:

I'm pleased to inform you that your manuscript has been deemed suitable for publication in PLOS ONE. Congratulations! Your manuscript is now with our production department. 

Kind regards, 

on behalf of

Dr. Susan Nittrouer 

Academic Editor

PLOS ONE